# REX: General-Purpose CNL with Code Generation Support

Adriano Carvalho 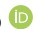

Centro Algoritmi, Universidade do Minho, 4800-058 Guimarães, Portugal; b4795@algoritmi.uminho.pt

**Abstract:** Controlled natural languages (CNLs) have been proposed to address some of the issues of natural language when it is used to express requirements. CNLs, however, are based on formal grammar, which can easily become complex, hard to read, and especially hard to write, and the implementation of support tools can also demand a significant effort. Moreover, unanticipated constructions cannot be handled or have to be handled in unexpected and cumbersome ways. In this article, we present REX, a CNL with a simple grammar that is, thus, easy to understand and easy to support, but still general purpose. To accomplish this, instead of trying to support every conceivable construction and imposing a language on the users, through a small but comprehensive set of rules and through patterns, users specify their own language and how natural it is. Another of the benefits of CNLs is the possibility to automate the transformation of a text or specification into something useful, thereby reducing manual labor and transformation errors. In this article, we also present the support tools that were used to transform a REX text into code and a complete application. It is also shown that this CNL and its support tools can be easily adapted to suit different needs.

**Keywords:** controlled natural language; code generation; programming; requirements specification





## 1. Introduction

The estimated cost of fixing an error during the requirements phase has been estimated as up to 1500 times cheaper than fixing an error in other phases of the project life-cycle [1]. One of the approaches proposed for the reduction and early detection of errors in the requirements is the use of controlled natural languages (CNLs) for writing requirements [2,3].

A CNL should read like natural language, such that it is accessible to all or most stakeholders. However, according to a formal grammar, there is always one interpretation for a given sentence [4]. Hence, with a CNL, it is possible to remove ambiguity and, through appropriate support tools, the reduction and early detection of errors in the requirements. CNLs, nevertheless, are not without limitations. They require readers, but especially writers, to know the grammar and learn to express themselves using that grammar in a way that does not impair their creativity [5]. Moreover, formal grammars can easily become complex, and the implementation of support tools demand a significant effort. Lastly, unanticipated constructions or patterns cannot be handled or have to be handled in unexpected and cumbersome ways. These limitations lead to trade-offs (e.g., comprehensive grammar vs. available support tools), which ultimately result in CNLs that are either limited in scope or not very useful. In the end, the CNLs that are found in the literature have at least one of the following limitations: not suitable or very limited for requirements specifications, no runtime code generation support, or no balance between natural and technical modes of expression.

In this article, we present REX (the implementation used for this article can be found at: https://github.com/b4795/REX, accessed on 24 June 2022), a CNL which distinguishes itself by having a small but comprehensive set of rules, and thus, a grammar that is simple, easy to understand, and easy to support. Despite the small set of rules, it is a general-purpose language because it can be extended by the users to suit their particular needs through patterns. In this article, code generation from REX is also demonstrated. Code

generation (as well as the generation of any other artifacts) is possible because of the CNL's formal grammar, bringing a number of benefits to add to those already mentioned.

This article is structured as follows. Section 2 provides some background and reviews of related work. Section 3 presents REX through four simple examples of increasing complexity, while Section 4 describes the implementation of support tools. This article ends with some conclusions and additional remarks in Section 5.

## 2. Background and Related Work

A requirement has been defined as "a statement that identifies a necessary attribute, capability, characteristic, or quality of a system in order for it to have value and utility to a user" [6]. Requirements are often the basis for communication between stakeholders with different backgrounds (e.g., users, developers, project managers), and so, requirements should be accessible and, whenever possible, easy to understand. There are different ways of documenting requirements. By far, the most common way is to use text, but there are also graphical notations (which end up relying on text as well). Text-based requirements are written, more often than not, using informal natural language, often ambiguous and thus, error-prone. Graphical notations introduce some formalisms, which help to reduce ambiguity, but there is still room for informality and, most of all, require stakeholders to know those notations. In some cases (e.g., for critical systems), formal methods are sometimes used, but these rely on languages and methods accessible to only a very few. Altogether, this means that, in practice, requirement specifications contain ambiguities that lead to costly errors.

The estimated cost of fixing an error during the requirements phase has been estimated as up to 1500 times cheaper than fixing an error in other phases of the project life-cycle [1]. Errors in the requirements are inevitable, hence, those who are able to detect and correct them early, such that they do not propagate, have a significant competitive advantage. Besides the cost of correcting errors, the cost of changing requirements is also high (an increasing necessity nowadays in order to adapt to fast changing business needs). There are, however, not only monetary and time-related costs, but also a psychological cost on the developers and other stakeholders, leading to a resistance to change and innovation.

We identify two major kinds of errors in the requirements: (1) errors when expressing or understanding requirements (we will call these verification errors); (2) errors estimating the realization value of requirements in practice (we will call these validation errors).

In the literature, we also find two major kinds of approaches which propose the reduction or early detection of errors in the requirements, namely: (1) new processes, and (2) new tools. If processes were followed scrupulously, most (if not all) errors would be found. The issue, in our opinion, is that typical human beings are unable to maintain the necessary level of focus and concentration required to follow processes that are often complex, have high overhead, and lack adequate support tools. This is exacerbated by the large size of typical requirement specifications. It is impossible for a typical human being to reason on such large texts with the necessary precision. In terms of tools, and to workaround the current limitations in natural language processing, controlled natural languages (CNLs) have been proposed for writing requirements [2,3].

A CNL should read like natural language, such that it is accessible to all or most stakeholders. However, according to a formal grammar, there is always one interpretation for a given sentence [4]. Hence, with a CNL, it is possible to remove ambiguity and, through appropriate support tools, the reduction and early detection of verification errors in the requirements. CNLs, nevertheless, are not without limitations. They require readers, and especially writers, to know the grammar and to learn to express themselves using that grammar in a way that does not impair their creativity [5]. Formal grammars can easily become complex, and the implementation of support tools demand a significant effort. Moreover, unanticipated constructions or patterns cannot be handled or have to be handled in unexpected and cumbersome ways. These limitations lead to trade-offs (e.g.,

comprehensive grammar vs. support tools) which ultimately result in CNLs that are either limited in scope or not very useful.

Besides removing ambiguity, another major benefit of using a CNL is that relevant artifacts can be predictably generated. This enables, in particular, code generation, automated verification, early validation, and in the end, reduced time-to-market. A requirements specification written in a CNL can be compiled into a formal model that can be verified to prove that there are no contradictions, that invariants are not violated, and more [2,3,7–10]. Similarly, a requirements specification can be compiled into prototypes or mockups, or ideally, into runtime or production code, which can be delivered to QA as well as to end-users, enabling the realization of the requirements to be tested and validated in practice as soon as possible.

In the literature, we find that Osmosian [11], Logical English [12,13], requs [14,15], and Gherkin [16], are the CNLs that most closely resemble REX. First, Osmosian is accompanied by compiler, desktop, editor, etc., all of them written in "plain english". This project seems to have been developed to prove the point that "plain english programming" is feasible (we believe that requirements can be regarded as just another programming language, yet one with a very high-level of abstraction and all-powerful), and thus, resorts almost completely in "plain english", which for practical purposes is not always appropriate. For practical purposes, there should be a balance between natural and technical modes of expression. Second, requs, like REX, has also been developed for runtime code generation, but from very limited use case specifications. Third, Cucumber is focused on acceptance tests specifications, which some may regard as a requirements specification; however, so far, it has not been sufficient for runtime code generation (as opposed to test code generation). Lastly, Logical English, developed in Prolog, with a complex and imposing grammar has also not been developed for runtime code generation.

In the literature, many other CNLs can be found, of which we highlight: (1) the CNLs presented by Silva et al. [17], which impose a strict and thus, limiting grammar; (2) SBVR [18], or SBVR-SE, to be more specific, and RuleSpeak [19] (supported by RuleXpress [20]) for managing business rules; and (3) ACE [4] and PENG [21] for knowledge representation using first-order logic.

In the end, the CNLs that are found in the literature have at least one of the following limitations: not suitable or very limited for requirements specifications, no runtime code generation support, or no balance between natural and technical modes of expression. In this article, we present REX, a CNL which distinguishes itself by having a small but comprehensive set of rules, and thus, a grammar that is simple, easy to understand, and easy to support. In this way, and as explained in the following sections, the limitations above do not apply to REX.

Looking back through a CNL for requirement specification, the two major kinds of requirement errors identified above are addressed as follows:

- Verification errors: ambiguity is removed (through a formal grammar), and automated verification is added (through code generation);
- Validation errors: the time/effort required to measure the realization value of requirements in practice is reduced (through code generation).

Altogether, some of the error-prone parts of requirements engineering can be automated (e.g., verification) and accelerated (e.g., validation), leading to reduced time-to-market, and enabling stakeholders to focus more of their time, energy, and motivation on producing value.

## 3. REX

### 3.1. Introduction

REX is a CNL that distinguishes itself by having a small but comprehensive set of rules (seven statements or phrases), and thus, a very simple grammar. This means that it has a low entry barrier since there is only a small set of rules to learn, and it is easy to support. Despite the small set of rules, as will be shown, it is a general purpose language because

it can be extended by the users to suit their needs. The rules that are defined should be natural and accessible to people from all backgrounds. The seven statements or phrases are:

- Class declaration (one statement);
- Class composition (one statement);
- Subclassing (one statement);
- Function/pattern declaration, definition, and call (two statements, one phrase);
- Class/object instantiation (one phrase).

Instead of trying to support every conceivable construction and instead of imposing a language on the users, and thus, a complex grammar and implementation, through the small set of rules that are available, the users can specify their own language using patterns. Since it is the users that define the language it is up to them how natural it is, whether it is concise or verbose, whether to enforce good practices (e.g., (Big) EARS patterns [22,23]), and so on. Moreover, when it comes to legacy system, patterns provide the flexibility needed to support a language that is already in use. Patterns are easy to understand and are precise, but they can be tedious and labor-intensive to work with since all patterns have to be specified *ipsis verbis* or close before being used. After some time, however, the most common and useful patterns have been specified and not much work should be needed, at least when it comes to specifying patterns. Despite having chosen patterns, we believe that the ideal is a combination of approaches. We envisage a specification that is parsed using both pattern recognition and formal grammars, depending on the context. For those areas for which a formal grammar has already been defined, we can leverage a workaround for the limitations of patterns.

Because REX relies on user-defined patterns, it attributes no semantics. That is also controlled by the users. The users can attribute semantics to a pattern by specifying it in terms of other patterns, or, as will be shown, the users can attribute semantics during the transformation to code or directly in code. In the end, a REX text can be seen as a tree whose leaves have to be defined elsewhere.

### 3.2. Hello, World!

The goal of the example in this section is to print "Hello, World!" on a console. This example's REX text is shown in Listing 1. It consists of two sentences, where each sentence ends with a period.

**Listing 1.** REX text for the "Hello, World!" example.

```
it_is_possible_to print_hello_world.
if launch_application, then print_hello_world.
```

The first sentence declares a pattern or a function (we will use the term function onwards because it provides a better definition for the ways in which a pattern can be used). Functions can be called or, in other words, used in the definition of other functions. The semantics of calling or using a function are not defined by the CNL; that is up to the users. However, in REX (and thus, in all of the examples in this article), it is set up to be the typical function call seen in imperative languages. The function's signature is defined by what is after the keyword (e.g., *print_hello_world*), and it can be used to call this function, as in the second sentence. The grammar (simplified) for a function declaration is shown in Listing 2. It is a limitation of the current implementation that *WORDs* have to be separated by one underscore and not spaces, as would be expected. The semantics or the implementation of this function are not specified here, as mentioned earlier. Later, that will be set to print "hello, world!" on a console, as expected. In REX, the first sentence is

transformed into a method named *print_hello_world* with no arguments, no return value, and no implementation, as will be shown.

**Listing 2.** Grammar for the function declaration (first sentence) in the "Hello, World!" example (simplified).

```
FunDecl     <-- "it_is_possible_to" FunDeclElem+ "."
FunDeclElem <-- WORD

Examples:
it_is_possible_to show_UI.
it_is_possible_to create_main_window.
it_is_possible_to add_account.
it_is_possible_to translate_to_portuguese.
```

The second sentence defines a function, including its signature, and a body (i.e., a list of function calls or just one). The grammar (simplified) for a function definition is shown in Listing 3. In REX, the semantics are such that whenever the function being defined is called (through its signature), the functions specified in the body are also called, in the same order as they have been specified. This, however, is enforced by REX, not by the CNL. In REX, the second sentence is transformed into a method named *launch_application* (no arguments, no return value), whose body is a call to a method named *print_hello_world*, as will be shown.

**Listing 3.** Grammar for the function definition (second sentence) in the "Hello, World!" example (simplified).

```
FunDef      <-- "if" FunDeclElem+ ","
"then" FunCallElem+ "."
FunDeclElem <-- WORD
FunCallElem <-- FunDeclElem

Examples:
if show_UI, then create_main_window.
if add_account_button_clicked, then add_account.
If translate_button_clicked, then translate_to_portuguese.
```

In REX, without user intervention, this text's model or AST (see Figure 1) is compiled into an Ecore model (see Figure 2), which is then used by EMF (Eclipse Modeling Framework) [24] to generate Java code (see Listing 4). Please refer to Section 4 for more details. At this point, no semantics or implementation were provided for *print_hello_world*, and as such, by default, EMF throws an exception, as shown in Listing 4. There are at least two alternatives to provide an implementation for *print_hello_world*: (1) update the method's implementation in the generated code file (EMF supports this, and is able to keep code that has been entered manually); (2) override the method in a subclass. Throughout this article, the second alternative will be followed. With this in mind, in Listing 5, the implementation of *print_hello_world* is set, alongside a possible main method.

```
platform:/resource/HelloWorld/src/hello.mydsl
  Model
    Operation Declarion Header Stmt
      Operation Parameter Connection Phrase print_hello_world
    Operation Definition Stmt
      Operation Declarion Header Stmt
        Operation Parameter Connection Phrase launch_application
      Operation Definition Body Stmts
        Operation Definition Body Function Call Phrase
          Operation Definition Body Function Call In Refs Ref Phrase
```

**Figure 1.** AST of the REX text in the "Hello, World!" example.

```
platform:/resource/HelloWorld/src/hello.ecore
  System
    _ops
      print_hello_world()
      launch_application()
        GenModel
```

**Figure 2.** Ecore model compiled from the REX text in the "Hello, World!" example.

**Listing 4.** Java code generated by EMF from the Ecore model compiled from the REX text in the "Hello, World!" example (simplified).

```java
public class _ops {

public void print_hello_world() {
throw new UnsupportedOperationException();
}

public void launch_application() {
print_hello_world();
}
}
```

**Listing 5.** Java code entered manually for the "Hello, World!" example (simplified).

```java
public class HelloApp extends _ops {

public static void main(String[] args) {
HelloApp hello = new HelloApp();
hello.launch_application();
}

public void print_hello_world() {
System.out.println(''Hello, World!'');
}
}
```

### 3.3. Hello, World! 2

The goal in this section is still to print "Hello, World!" on a console. However, what is printed is set directly on the REX code, instead of being hardcoded in the Java code. This example's REX text is shown in Listing 6, and it consists of three sentences.

The first sentence declares the existence of a class of objects (e.g., strings), through the name of that class of objects in plural and singular forms. Objects can be created (or

instantiated), modified, and passed around. Objects can also hold other objects, through class composition (e.g., *"a name is_composed_by a first_name, and a last_name."*, *"a text is_composed_by zero_or_more strings."*), which has not been used in any of the examples. Moreover, as will be shown for the second sentence, certain parts of a function signature can be restricted to only accept references to objects of a particular class and its subclasses (possible through subclassing, e.g., *"a first_name is a string."*, which has also not been used in any of the examples). For examples of class composition and subclassing, see: https://github.com/b4795/REX/blob/main/examples/Bank/src/bank.mydsl, accessed on 24 June 2022. When a function accepts references to objects, the corresponding function's body, when there is one, can reference those objects and pass them to other functions (see the next section for an example). The grammar (simplified) for a class declaration is shown in Listing 7.

**Listing 6.** REX text for the "Hello, World! 2" example.

```
there_are strings (string).
it_is_possible_to print a string S.
if launch_application, then print ``Hello, World!''.
```

**Listing 7.** Grammar for the class declaration (first sentence) in the "Hello, World! 2" example (simplified).

```
ClassDecl <-- "there_are" ClassPlu "(" ClassSin ")" "."
ClassPlu  <-- WORD
ClassSin  <-- WORD

Examples:
there_are persons (person).
There_are names (name).
there_are clients (client).
there_are banks (bank).
```

The semantics behind the class of objects that is declared are also not specified here. That will depend on the way the objects are used, to which functions they are passed, and how those functions are defined. In REX, however, strings, alongside booleans and integers, are considered built-in, and some semantics are predefined (e.g., built-in objects cannot hold other objects). It is a limitation of the current implementation that their existence has to be declared explicitly. The predefined semantics of a string include: (1) a string is an ordered set of characters, and (2) a string object can be instantiated with characters between double quotes, as shown in the third sentence. In REX, the first sentence is transformed into a class *_string* with an attribute named "value", whose type is Java's built-in String, as well as one setter and one getter, as shown in Listing 8.

**Listing 8.** Java code, regarding the string class, generated by EMF from the Ecore model compiled from the REX text in the "Hello, World! 2" example (simplified).

```
public class _string {
protected String value;

public String getValue() {
return value;
}

public void setValue(String newValue) {
value = newValue;
}
}
```

The second sentence, as in the previous section, declares a function, but, as mentioned above, part of the function signature is restricted to only accept references to a particular class of objects (e.g., strings). The only reference is also given the name "S". An updated grammar (still simplified) for a function declaration is shown in Listing 9. As before, semantics are not specified here. Later, that will be set to print the value of the string S on a console. In REX, the second sentence is transformed into a method named *print__string* with one argument of type *_string*, no return value, and no implementation, as will be shown.

**Listing 9.** Grammar for the function declaration (second sentence) in the "Hello, World! 2" example (simplified).

```
FunDecl     <-- "it_is_possible_to" FunDeclElem+ "."
FunDeclElem <-- WORD
| (("a" | "an") ClassSin)

Examples:
it_is_possible_to show a UI X.
it_is_possible_to show a widget W.
it_is_possible_to add an account ACC.
```

The third sentence, as in the previous section, defines a function. An updated grammar (still simplified) for a function definition is shown in Listing 10. In REX, the semantics are the same as in the previous section, but, in this case, a string object is instantiated (implicitly) with the value "Hello, World!" and a reference to that object is passed to the function *print__string*, as will be shown.

**Listing 10.** Grammar for the function definition (third sentence) in the "Hello, World! 2" example (simplified).

```
FunDef      <-- "if" FunDeclElem+ ","
"then"  FunCallElem+ "."
FunDeclElem <-- WORD
FunCallElem <-- WORD
| STRING

Examples:
if show UI, then create main window named "App".
if show UI, then add account named "savings".
If translate button clicked, then
translate "hello" to portuguese.
```

In REX, the text in Listing 6 is compiled into the Ecore model shown in Figure 3, which is then used by EMF to generate the Java code shown in Listing 11. Lastly, in Listing 12, the semantics of *print a string S* are set, alongside a possible main method.

**Figure 3.** Ecore model compiled from the REX text in the "Hello, World! 2" example.

**Listing 11.** Java code generated by EMF from the Ecore model compiled from the REX text in the "Hello, World! 2" example (simplified).

```
public class _ops {

public void print__string(_string S) {
throw new UnsupportedOperationException();
}

public void launch_application() {
SystemFactory factory = new SystemFactory();
_string _local__string0 = factory.create_string();
_local__string0.setValue('`Hello, World!'');
print__string(_local__string0);
}
}
```

**Listing 12.** Java code entered manually for the "Hello, World! 2" example (simplified).

```
public class Hello2App extends _ops {

public static void main(String[] args) {
Hello2App hello2 = new Hello2App();
hello2.launch_application();
}

public void print__string(_string S) {
System.out.println(S.getValue());
}
}
```

### 3.4. Hello, World! 3

In this section, the goal is to, first, ask the user's name, and then, to great the user using the name provided. This example's REX text is shown in Listing 13. It consists of five sentences, where the fifth sentence is composed of nine phrases.

**Listing 13.** REX text for the "Hello, World! 3" example.

```
there_are strings (string).

it_is_possible_to print a string S.
it_is_possible_to start_a_new_line.
it_is_possible_to read_input_from_user_and_store_it_in a string S.

if launch_application, then
there_is_one string S,
print "Hey! What's your name?",
start_a_new_line,
read_input_from_user_and_store_it_in S,
print "Hello ",
print S,
print "! Welcome!",
start_a_new_line.
```

The implications of sentences one to four should be clear by now. The semantics of the forth sentence, however, will be set such that the reference S works as an output or return value (instead of as an input, as in the previous section). In REX, all objects are passed by reference, and thus, they can be both inputs and outputs. Whether a reference works as an input or output, or both, depends on the functions that are called, and their implementation. That, however, should be clear from the function signature, as in this example.

Regarding the fifth sentence, with the exception of the second phrase, the implications of all phrases should also be clear by now. In the second phrase, an object of the specified class is instantiated (explicitly) such that it can hold a value and such that it can be referenced by the specified name. Here, it is instantiated an object of the class string, named S, which is then used to hold the user's input (phrases 5 and 7). The grammar (simplified) for a class instantiation is shown in Listing 14.

**Listing 14.** Grammar for the class instantiation (third sentence, second phrase) in the "Hello, World! 3" example (simplified).

```
ClassInst <-- "there_is_one" ClassSin WORD "."
ClassSin  <-- WORD

Examples:
there_is_one person ADA.
There_is_one name N.
there_is_one client C.
```

In REX, the text in Listing 13 is compiled into the Ecore model shown in Figure 4, which leads to the generation of the Java code shown in Listing 15 (only the *launch_application* method is shown since the rest should be straightforward by now). In Listing 16, the semantics of undefined functions are set, alongside a possible main method.

**Figure 4.** Ecore model compiled from the REX text in the "Hello, World! 3" example.

**Listing 15.** Java code generated by EMF from the Ecore model compiled from the REX text in the "Hello, World! 3" example (simplified, and only the *launch_application* method is shown).

```
public void launch_application() {
SystemFactory factory = new SystemFactory();
_string _local__string0 = factory.create_string();
_local__string0.setValue("Hey! What's your name?");
_string _local__string1 = factory.create_string();
_local__string1.setValue("Hello ");
_string _local__string2 = factory.create_string();
_local__string2.setValue("! Welcome!");
_string S = factory.create_string();
print__string(_local__string0);
start_a_new_line();
read_input_from_user_and_store_it_in__string(S);
print__string(_local__string1);
print__string(S);
print__string(_local__string2);
start_a_new_line();
}
```

**Listing 16.** Java code entered manually for the "Hello, World! 3" example (simplified).

```java
public class Hello3App extends _ops {

public static void main(String[] args) {
Hello3App hello3 = new Hello3App();
hello3.launch_application();
}

public void print__string(_string S) {
System.out.print(S.getValue()); // no new line.
}

public void start_a_new_line() {
System.out.println(); // new line.
}

public void read_input_from_user_and_store_it_in__string(_string S) {
Scanner scanner = new Scanner(System.in);
S.setValue(scanner.nextLine());
}
}
```

*3.5. Fibonacci*

The goal here is to calculate the result of the application of the Fibonacci's function to an integer N or, in other words, to calculate the Nth element in the Fibonacci sequence. This example's REX text is shown in Listing 17. Mathematical arithmetic and comparison are verbose. This is another limitation of the current implementation. Nevertheless, nothing precludes including mathematical operators in function signatures (e.g., *an integer A > an integers B is a boolean*). The only thing that is needed is to treat those operators like any other alphanumeric character. Most of the text should be clear by now, however, conditions are introduced and these need some background. In REX, functions with signatures ending with *is a boolean* can be used as conditions. We recognize this construction is a bit unexpected. It was chosen because it simplifies the implementation, but completely different approaches are possible. For example, the declaration "*it_is_possible_that an integer X is_or_is_not_greater_than_or_equal_to an integer Y.*", because of the prefix *is_or_is_not_* could imply a boolean return value, as well two different function signatures, namely: "*…is_greater_than_or_equal_to …*" and "*…is_not_greater_than_or_equal_to …*". It should also be noted that: "*an integer X is an integer Y*" is expected to correspond and will be set to the assignment of Y's value to X.

**Listing 17.** REX text for the "Fibonacci" example.

```
there_are booleans (boolean).
there_are integers (integer).

it_is_possible_that an integer X is an integer Y.
it_is_possible_that an integer X
is_greater_than_or_equal_to
an integer Y
is a boolean B.

it_is_possible the_addition_of an integer A
with an integer B
is an integer X.
it_is_possible the_difference_between an integer A
and an integer B
is an integer X.

it_is_possible_to output an integer X.

if the_fibonacci_function_of an integer X is an integer Y, then
if X is_greater_than_or_equal_to 3, then
there_is_one integer N1,
there_is_one integer N2,
the_difference_between X and 1 is N1,
the_difference_between X and 2 is N2,
there_is_one integer A,
there_is_one integer B,
the_fibonacci_function_of N1 is A,
the_fibonacci_function_of N2 is B,
the_addition_of A with B is Y;
else
Y is 1.

if launch application, then
there_is_one integer A,
there_is_one integer B,
there_is_one integer C,
the_fibonacci_function_of 1 is A,
the_fibonacci_function_of 2 is B,
the_fibonacci_function_of 10 is C,
output A,
output B,
output C.
```

In REX, the text in Listing 17 is compiled into the Ecore model shown in Figure 5. In Listing 18, the semantics of undefined functions are set, alongside a possible main method.

**Figure 5.** Ecore model compiled from the REX text in the "Fibonacci" example.

**Listing 18.** Java code entered manually for the "Fibonacci" example (simplified).

```
public class FibApp extends _ops {

public static void main(String[] args) {
FibApp fib = new FibApp();
fib.launch_application();
}

public void _integer_is__integer(_integer X, _integer Y) {
X.setValue(Y.getValue());;
}

public void _integer_is_greater_than_or_equal_to__integer_is__boolean(
_integer X, _integer Y, _boolean B) {
B.setValue(X.getValue() >= Y.getValue());
}

public void the_addition_of__integer_with__integer_is__integer(
_integer A, _integer B, _integer X) {
X.setValue(A.getValue() + B.getValue());
}

public void the_difference_between__integer_and__integer_is__integer(
_integer A, _integer B, _integer X) {
X.setValue(A.getValue() - B.getValue());
}

public void output__integer(_integer X) {
System.out.println(X.getValue());
}
}
```

## 4. Implementation

In this section, the implementation of REX using the Eclipse IDE is described. Because REX has a small (but comprehensive) set of rules, and thus, a very simple grammar, its implementation is also quite straightforward. Using the tools provided by the Eclipse IDE, only two REX-specific translation units (or source files) are needed, namely, one grammar (44 rules, 113 lines of code, empty lines not included, approx. 2.5 lines of code per rule) (see https://github.com/b4795/REX/blob/main/src/org.xtext.example.mydsl/src/org/

xtext/example/mydsl/MyDsl.xtext, accessed on 24 June 2022), and one model transformation (515 lines of code) (see https://github.com/b4795/REX/blob/main/src/org.xtext.example.mydsl/src/org/xtext/example/mydsl/generator/mydsl2ecore.etl, accessed on 24 June 2022). More specifically, we have used Eclipse IDE for Java and DSL Developers, version 2021-09 [25], with Xtext 2.25.0 [26], Epsilon 2.3.0 [27], EMF 2.27.0 [24], and JDK 3.18.900 (builtin).

The implementation was performed on one Eclipse IDE instance, which we will call "Eclipse-dev". As illustrated in Figure 6, we have entered REX's grammar onto Eclipse-dev and used Xtext to generate a corresponding metamodel and language editor. With the code generated by Xtext, we start a new Eclipse instance, which we will call "Eclipse-usr", featuring, most of all, the REX language editor.

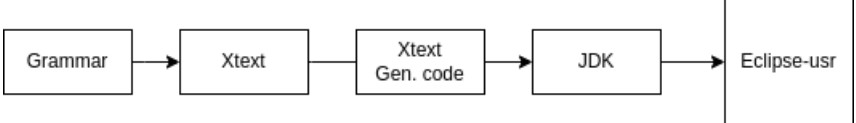

**Figure 6.** Implementation of REX in Eclipse-dev.

For each of the examples presented in the previous section, we have entered the respective REX text onto the REX language editor in Eclipse-usr. As illustrated in Figure 7, whenever the REX text file is saved, Epsilon is triggered to generate a corresponding Ecore model, as explained in the previous section. To do this, Epsilon requires a model transformation script, and the model to transform. The model to transform is the REX text file, which is automatically transformed from a text representation into a model representation, whenever necessary. The model transformation script, on the other end, comes with the Eclipse-usr instance. To do that, the script was entered onto the Eclipse-dev and a configuration file set to include that file in the Eclipse-usr instance. With the Ecore model in place, Java code generation is triggered, using EMF, and it is now up to the users to define what has been left undefined, as explained in the previous section.

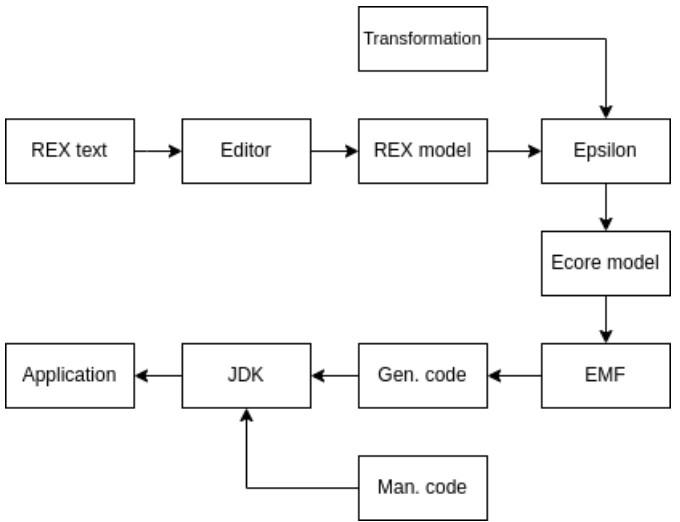

**Figure 7.** Implementation and usage of REX in Eclipse-usr.

## 5. Conclusions

In this article, REX has been presented, a general-purpose CNL with code generation support. REX distinguishes itself by having a small but comprehensive set of rules, and thus, a grammar that is simple, easy to understand, and easy to support. To do that, four examples of increasing complexity were given. After that, its implementation was also described.

Despite the examples shown in this article being very simple, we are unable to find reasons for it not being able to scale to much larger applications. Similarly, the examples in

this article restrict themselves to Java code generation and to very simple console-based applications. Nevertheless, Java is used here as just an example, and by changing the code generator and/or the REX text transformation, any other language could be supported, as well as any other type of application (e.g., GUI).

As mentioned throughout the article, REX and its current implementation have some limitations. Hence, as future work, we propose to address those already mentioned (i.e., *WORDs* have to be separated by one underscore and not spaces, the existence of built-in classes has to be declared explicitly, mathematical operators not supported), as well as to add new features, namely: references to functions and complexity management (e.g., namespaces). Besides, and more importantly, to evaluate REX, we propose to conduct some case studies and questionnaires in cooperation with industry partners.

The major purpose of REX and this article is to demonstrate that a general-purpose CNL, that is more than enough for many projects, and that is easy to support, is best achieved by letting the users specify their own language, instead of imposing a language on them (an advantage of itself). In this way, a simple grammar that is also easy to understand and support, is sufficient. Another major purpose of REX is to serve as a reference implementation, that can be modified to suit different needs, within practical or research efforts.

With REX, we believe it is possible that, for example, business experts use the CNL to specify what they need in a language they understand (and perhaps more importantly, in a language they created), while technical experts specify model transformations and some code to ensure the intended semantics are preserved through low-level abstractions. If it is important to the business experts, the CNL should be used; otherwise, that effort should be transferred to the technical experts, which may also pick the CNL to do part of the necessary work. A collaboration between business and technical experts is still necessary, but the role of business experts is much more prominent. Business experts participate more actively on the development and can directly affect the implementation, instead of indirectly through the technical experts and, in some cases, without having to cross multiple organization layers, which is especially advantageous when the business logic changes more often than low-level implementation details.

Lastly, besides business applications, we believe it is possible to make software development more accessible and empower less technical people. Moreover, the learning curve towards more traditional software development can be made less steep, and existing technical experts can benefit not only from increased productivity but also with access to languages and methods accessible to only a few (e.g., formal methods).

**Funding:** This research received no external funding.

**Institutional Review Board Statement:** Not applicable

**Informed Consent Statement:** Not applicable

**Data Availability Statement:** Not applicable

**Conflicts of Interest:** The author declares no conflict of interest.

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
