# Peer review of "REX: General-Purpose CNL with Code Generation Support"

_applsci, doi:10.3390/app12157700_

Round 1

Reviewer 1 Report

The authors presentet REX, a controlled natural languages with a simple grammar, easy to understand, easy to support, and general-purpose. 

Some improvement suggestions:

1) The authors do not present in the introduction,  why it is important to propose the REX, which is the main contribution of the work and the motivation for the research. I suggest this be added in the introduction. 

2) In the section on related work, it is important that the authors make it clear what is the main difference between this research and those existing in the literature.

3) The manuscript does not have a method section.

3.1 What methodology was adopted to develop this research?

3.2 What are the research questions?

3.3 How were they answered?

3.4 Why is proposing REX important?

The article does not present information and this should be added to the body of the manuscript.

4) What are the limitations of REX?

5) What are the threats to validity? I suggest authors add this to the manuscript as well as a discussion section.

Author Response

Dear Sir or Madam,

Thank you very much for the time spent carefully reviewing our articles.

Taking into account yours and other reviewers feedback we updated the article as follows:

  • Through the article: small wording changes as suggested by the reviewers.
  • Introduction: Explain the limitations in the literature. Make clearer what distinguishes REX from the CNLs in the literature.
  • Related work: Delete a repeated paragraph. Make clearer what distinguishes REX from the CNLs in the literature.

Since what is presented in this article did not follow a predefined methodology we have chosen to not include a "Methods" section. What is presented in this article is something that we stumble upon and no particular methodology was used to achieve this.

It is our understanding that the limitations of our work as well as threats to validity are included in the "Conclusion" section. It seems to us that separating these subjects into separate section as no significant advantage.

Similarly, our "Discussion" is included in the "Conclusion" section.

In this way, we finish our message to you, hoping that you will understand our choices.

Best regards,

Adriano Carvalho

Reviewer 2 Report

This manuscript reports on the development of a new general-purpose controlled natural language, REX, and supporting tools. REX is presented as a simple-to-use and simple-to-learn grammar with broad application and scope. The manuscript is dense reflecting on the substantial work that went into developing this new CNL grammar. This grammar enables code generation through the patterns that users can develop themselves, by extending the grammar to fit their own needs. The author provided a succinct list of alternative CNLs, and a list of disadvantages was also provided for them when compared to the proposed REX. The examples provided guide the reader through REX and illustrate the various rules (statements and phrases). 

There are still several limitations that prevent this CNL from reaching adoption and becoming useful. Namely, the fact that words need to be separated with underscore instead of spaces, lack of implicit classes and others such as missing operators. The work is already substantial but it feels that these limitations should be addressed before publishing this work, especially when claims are made about its advantages and benefits when compared to other existing CNLs. Perhaps a little out of the scope of this paper, but it would be good if the author could discuss the use of recent natural language machine learning techniques (BERT, etc.), which are increasingly used to tackle some of the issues reported here, requirements specifications, code generation and others. 

List of other suggestions to the author:

  • Consider revising the sentence from lines 4-5, to avoid using terms such as “awkwardly”. The same applies in the introduction (29-30).

  • Consider replacing “a CNL with only a few rules”, to something like “a CNL with a small comprehensive set of rules”.

  • Consider removing this paragraph with lines 100-105 as it reads repetitive and it is very similar to what has been written in the abstract and introduction.

  • Avoid using terms such as “among other things” (line 56-57) as it makes the statements vague and subjective. If the “other things” are not listed, it is preferable to avoid mentioning them altogether.

  • When comparing REX with other CNLs, consider rephrasing “not as natural as REX can be”. It would be better if the author describes the limitations of those CNLs and the advantages of REX, instead.

  • Consider rephrasing (line 177) “Others might say that they extend the domain of discourse.” as it is not very scientific or even appropriate.

  • EMF is not defined in the text. Please define it on the first appearance in the text.

  • Please consider adding line numbers mentioned in the text, to the code blocks as it improves readability dramatically.

Author Response

Dear Sir or Madam,

Thank you very much for the time spent carefully reviewing our articles.

Taking into account yours and other reviewers feedback we updated the article as follows:

  • Through the article: small wording changes as suggested by the reviewers.
  • Introduction: Explain the limitations in the literature. Make clearer what distinguishes REX from the CNLs in the literature.
  • Related work: Delete a repeated paragraph. Make clearer what distinguishes REX from the CNLs in the literature.

With this update, we believe we have addressed all your concerns.

In this way, we finish our message to you, hoping that you will understand our choices.

Best regards,

Adriano Carvalho

Reviewer 3 Report

In this paper, authors presented REX, a 5 CNL with a simple grammar that is easy to understand, easy to support, and general-purpose. They also presented the support tools that were used to transform a REX text into code and a complete application. Authors identifed two major kinds of errors in the requirements: (1) errors when expressing or understanding requirements (we will call these verification errors); (2) errors estimating the realization value of requirements in practice (we will call these validation errors).

Motivation/Objective of the study supported by related references should be included in the Introduction section. This is important for potential readers to appreciate significance of the study.

Related work section should be organized either in tabular forms or sub section wise so readers can appreciate in better way. Also relevant recent literature review should be further explore and added.

Discussion section should be included with related references mapping with implementation/results obtained.

Implications of this study for practitioners (along with practical benefits) and researchers should be included before Conclusion section.

Limitations and Future research directions should be extended and organize as sub sections of the Conclusion. 

Author Response

Dear Sir or Madam,

Thank you very much for the time spent carefully reviewing our articles.

Taking into account yours and other reviewers feedback we updated the article as follows:

  • Through the article: small wording changes as suggested by the reviewers.
  • Introduction: Explain the limitations in the literature. Make clearer what distinguishes REX from the CNLs in the literature.
  • Related work: Delete a repeated paragraph. Make clearer what distinguishes REX from the CNLs in the literature.

It is our understanding that the implications, the limitations, and future research directions of this work, are included in the "Conclusion" section. It seems to us that separating these subjects into separate section has no significant advantage.

Similarly, our "Discussion" is included in the "Conclusion" section.

In this way, we finish our message to you, hoping that you will understand our choices.

Best regards,

Adriano Carvalho

Round 2

Reviewer 1 Report

Congratulations to the authors for the corrections. In my opinion, the article is ready to be published.

Author Response

Dear Sir or Madam,

Thank you very much for the time spent carefully reviewing our article.

Taking into account your and other reviewers feedback we have updated the article as follows.

First, several typos and errors throughout the article have been corrected.

Second, the main takeway from this paper has been made clear. To do that, the "Conclusions" section has been modified to include the following:

The major purpose of REX and this article is to demonstrate that a general-purpose CNL, that is more than enough for many projects, and that is easy to support, is best achieved by letting the users specify their own language, instead of imposing a language on them (an advantage of itself). In this way, a simple grammar, that is also easy to understand and to support, is sufficient. Another major purpose of REX is to serve as a reference implementation, that can be modified to suit different needs, within practical or research efforts.

In this way, we finish our message to you, hoping that you will understand our choices.

Best regards,
Adriano Carvalho

Reviewer 2 Report

Dear authors,

Thanks for addressing the issues and comments reported. I would only recommend one last round of careful proofreading to identify possible typos and replace non-idiomatic expressions.

Best regards

Author Response

(The authors gave the same response as above.)

Reviewer 3 Report

Authors should include what is takeaway as implications of this study in the Conclusion section along with limitations and future research directions in explicit manner. These are very important for any such research study like an abstract/summary.

Author Response

Dear Sir or Madam,

Thank you very much for the time spent carefully reviewing our article.

Taking into account your and other reviewers feedback we have updated the article as follows.

First, several typos and errors throughout the article have been corrected.

Second, the main takeway from this paper has been made clear. To do that, the "Conclusions" section has been modified to include the following:

The major purpose of REX and this article is to demonstrate that a general-purpose CNL, that is more than enough for many projects, and that is easy to support, is best achieved by letting the users specify their own language, instead of imposing a language on them (an advantage of itself).  In this way, a simple grammar, that is also easy to understand and to support, is sufficient.  Another major purpose of REX is to serve as a reference implementation, that can be modified to suit different needs, within practical or research efforts.

Regarding your feedback regarding "limitations and future research directions" we believe we address your concerns in the "Conclusions" section with the following paragraph:

As mentioned throughout the article, REX and its current implementation have some limitations.  Hence, as future work we propose to address those already mentioned (i.e., WORDs have to be separated by one underscore and not spaces, the existence of built-in classes has to be declared explicitly, mathematical operators not supported) as well as to add new features, namely: references to functions, and complexity management (e.g., namespaces).  Besides, and more importantly, to evaluate REX we propose to conduct some case studies and questionnaires in cooperation with industry partners.

In this way, we finish our message to you, hoping that you will understand our choices.

Best regards,
Adriano Carvalho
